# Daptomycin Population Pharmacokinetics in Patients Affected by Severe Gram-Positive Infections: An Update

**DOI:** 10.3390/antibiotics11070914

**Published:** 2022-07-07

**Authors:** Giuseppe Balice, Claudio Passino, Maria Grazia Bongiorni, Luca Segreti, Alessandro Russo, Marianna Lastella, Giacomo Luci, Marco Falcone, Antonello Di Paolo

**Affiliations:** 1Sant’Anna School of Advanced Studies, Piazza Martiri della Libertà, 56127 Pisa, Italy; passino@ftgm.it; 2Hospices Civils de Lyon, Service Hospitalo-Universitaire de Pharmaco-Toxicologie, 162 Avenue Lacassagne, 69003 Lyon, France; 3Fondazione Toscana Gabriele Monasterio, Via Giuseppe Moruzzi 1, 56124 Pisa, Italy; 4Unit of Cardiovascular Diseases, Pisa University Hospital, Via Paradisa 2, 56100 Pisa, Italy; m.g.bongiorni@med.unipi.it (M.G.B.); l.segreti@ao-pisa.toscana.it (L.S.); 5Infectious and Tropical Disease Unit, Department of Medical and Surgical Sciences, “Magna Graecia” University of Catanzaro, Viale Europa, 88100 Catanzaro, Italy; a.russo@unicz.it; 6Department of Clinical and Experimental Medicine, University of Pisa, Via Savi 10, 56126 Pisa, Italy; marianna.lastella@unipi.it (M.L.); giacomo.luci@gmail.com (G.L.); antonello.dipaolo@unipi.it (A.D.P.); 7Unit of Clinical Pharmacology, Pisa University Hospital, Via Roma 55, 56126 Pisa, Italy; 8Unit of Infectious Diseases, Pisa University Hospital, Via Paradisa 2, 56100 Pisa, Italy; marco.falcone@unipi.it

**Keywords:** daptomycin, population pharmacokinetics, simulation, efficacy, toxicity

## Abstract

Daptomycin pharmacokinetics may not depend on renal function only and it significantly differs between healthy volunteers and severely ill patients. Herein, we propose a population pharmacokinetics model based on 424 plasma daptomycin concentrations collected from 156 patients affected by severe Gram-positive infections during a routine therapeutic drug monitoring protocol. Model building and validation were performed using NONMEM 7.2 (ICON plc), Xpose4 and Perl-speaks-to-NONMEM. The final pop-PK model was a one-compartment first-order elimination model, with a 2.7% IIV for drug clearance (Cl), influence of creatinine clearance on drug clearance and of sex on distribution volume. After model validation, we simulated 10,000 patients with the Monte-Carlo method to predict the efficacy and tolerability of different daptomycin daily dosages. For the most common 6 mg/kg daily dose, the simulated probability of overcoming the toxic minimum concentration (24.3 mg/L) was 14.8% and the efficacy (expressed as a cumulative fraction of response) against methicillin-resistant *S. aureus*, *S. pneumoniae* and *E. faecium* was 95.77%, 99.99% and 68%, respectively. According to the model-informed precision dosing paradigm, pharmacokinetic models such as ours could help clinicians to perform patient-tailored antimicrobial dosing and maximize the odds of therapy success without neglecting toxicity risks.

## 1. Introduction

Daptomycin is a lipopeptide compound with concentration-dependent anti-bacterial properties [1], which has proven to be non-inferior to the standard-of-care for several Gram-positive bacterial infections. In some cases, such as in the presence of vancomycin-resistant strains, it is the first line of treatment. In particular, daptomycin is authorized in Europe for the treatment of *S. aureus* blood-stream infections (BSI), right-sided infective endocarditis (IE) and complicated skin and soft tissue infections (cSSTI) in adults at recommended doses of 6, 6 and 4 mg/kg, respectively [2]. In pediatric patients, it is authorized for BSI and cSSTI treatment at age-adjusted dosages. Besides such official indications, several other Gram-positive infections are currently treated with off-label daptomycin, as Ye and colleagues recently synthesized [3].

Pre-registration pharmacokinetic analyses, such as the one performed by Dvorchik et al. on 24 healthy volunteers aged between 21 and 45 years [4], showed that daptomycin is found mainly unmodified in urines; thus, its clearance (Cl) mainly relies on renal function. Nonetheless, since daptomycin authorization in 2003, further clinical studies have pointed out new and more complex aspects of its pharmacokinetics, especially regarding severely ill patients. Falcone and colleagues extensively studied the impact of renal function on pharmacokinetics in 35 patients affected by severe Gram-positive infection and treated with daptomycin 6–8 mg/kg [5]. Their findings were partially consistent with the expectations, since patients with a plasma creatinine clearance (CrCl) of <40 mL/min had significantly lower daptomycin Cl than the others, but a significant statistical correlation between these two measures was lacking. Bubalo and colleagues studied the pharmacokinetics of daptomycin 6 mg/kg in 29 adults affected by neutropenic fever [6], obtaining results in conflict with the previous literature. Daptomycin Cl in their patients was not correlated with CrCl, with no difference in the calculation formula, and it was higher than that assessed in Phase I studies, as well as the volume of distribution at steady-state (V_ss_), which in turn accounted for lower maximum plasma concentrations (C_max_) than assessed before.

Taken together, those elements showed that adapting the dose on CrCl alone was not sufficient to ensure the most effective and less toxic daptomycin course in all “real-life” situations. Falcone and colleagues, thus, suggested to systematically introduce therapeutic drug monitoring (TDM) for daptomycin [7]. To complement this technique, population pharmacokinetics (pop-PK) approaches may be useful to initially guide clinicians in regimen choice, by considering multiple patient-related factors at a time.

The first proposed pop-PK model, developed by Dvorchik and colleagues, embedded a two-compartment first-order structural model with a covariate model including CrCl, dialysis status, body temperature, body weight, sex and infectious status [8]. By including pharmacokinetics data of 282 individuals, either healthy or affected by Gram-positive bacterial infections, enrolled in Phase I to III clinical trials, it remained the largest analysis for several years after its publication. Ten years later, Chaves and colleagues updated Dvorchik’s model, adding data of patients enrolled in five more Phase I to IV trials, to reach a total of 442 analyzed patients [9]. Their new covariate model extended the original one, by including a two-level covariate for dialysis membrane type and a five-level covariate for the final diagnosis. Xu and colleagues further refined this model, by adding 17 more patients enrolled in other studies, to better understand daptomycin pharmacokinetics in patients under continuous renal replacement therapy (CRRT) [10].

Other models have been proposed, based on smaller yet more precise datasets, in line with the concept that larger and all-encompassing datasets may introduce unnecessary noise. Xie et al. pooled 32 patients from 4 previous studies to address the unsolved question of which is the best daptomycin regimen for CRRT patients [11], while Soraluce et al. enrolled 16 patients hospitalized in their intensive care unit (ICU), 4 of whom underwent CRRT, to assess the effects of their profound pathophysiological changes on daptomycin pharmacokinetics [12]. Di Paolo and colleagues enrolled 58 patients with severe Gram-positive infections receiving daptomycin doses ranging from 4 to 12 mg/kg as 30-min infusions [13]. After correction for covariates, the authors claimed that, on the same dataset, their model predicted lower C_max_ and higher volumes of distribution (V_d_) than those calculated with previous models. Notably, pharmacokinetics of daptomycin was found to be time-dependent, as a consequence of the rapidly changing pathophysiological conditions of patients [14].

Therefore, the main aim of the present work was to upgrade and revalidate the model initially proposed by Di Paolo et al. for patients affected by severe Gram-positive infections, through the addition of new patients to its original dataset. The secondary aim was to reassess the tolerability and efficacy of different daptomycin dosages by simulations based on the Monte-Carlo method.

## 2. Results

### 2.1. Enrolled Patients

Synthetic patients’ data are provided in Table 1 and the inclusion flowchart is provided in Figure 1. Of note, cohorts for model building and model validation are adequately balanced for all the covariates except sex, thus confirming the efficacy of the random subsampling algorithm used. The twenty-two patients included in the second round did not introduce any substantial inhomogeneity in the dataset. Unbalance in the sex ratio is noticeable in all the study subgroups.

### 2.2. Daptomycin Plasma Concentrations

Measured daptomycin concentrations are reported in Table 2. At a first look, the mean peak concentration for the 10 mg/kg dose seems lower than expected, and conversely, the mean trough concentration seems higher. Nonetheless, the 6 mg/kg daily dose was the most prescribed in our hospital, and conversely, the 10 mg/kg and 12 mg/kg were the least frequently administered. Thus, peak and trough concentrations recorded for these dose classes may have been more susceptible to random sampling fluctuation and should be interpreted with caution.

### 2.3. Pop-PK Model Development

After 52 subsequent refinement steps, in which the OFV has been reduced by 264 units (from initial 1847.936 to final 1583.887), the final pop-PK model is arranged as in the following equations:(1)$$Cl\;\left(l/h\right)=\theta_{1}+\theta_{5}\times \frac{\left(Cl_{Cr}-63.35\right)}{100}+\eta$$
and
(2)$$V_{d}\;\left(l\right)=\theta_{2}-\theta_{6}\times \left(Sex-0.309\right)$$

*Sex* was coded as 0 for males and 1 for females. Parameter estimates are reported in Table 3, and goodness-of-fit plots are showed in Figure 2. In the graphical analysis, the model seems to overestimate plasma concentrations between the two sampling times (Figure 2D), although the correlation between observations and individual predictions is good.

The basic model is composed of a one-compartment first-order elimination structural model, with an IIV for Cl, and a complex-shaped residual error model. For individual predictions (iPRED) lower than 80 mg/L, the residual error is calculated with a slope-intercept model dependent on additive and proportional parameter errors, while for higher iPRED values, the model becomes proportional to a lone third parameter (Table 3). *η*- and an *ε-shrinkage* were 19.9% and 12.4%, respectively, which are both under the goodness reference value of 20%. The model successfully passed the internal validation, based on a 500-samples VPC (Figure 3), a 1000-samples NPC (Figure 4A) and a 1500-samples bootstrap (Table 3).

The external validation was performed running the model using the validation dataset. Because of the reduced size of the dataset, it was not possible to execute a VPC, but the NPC coverage plot (Figure 4B) ruled out abnormal trends in the DV/iPRED ratio.

The final model was used to predict the daptomycin plasma concentrations in 10,000 patients simulated through the Monte-Carlo method. The results are presented in Figure 5 and Table 4. Interestingly, in our simulation, fixed doses performed better than weight-adjusted doses. For instance, a fixed dose of 550 mg/day had 21.50% and 78.60% chance of resulting in trough and peak concentrations over the cutoffs, performing better than the daily dose of 8–9 mg/kg, which had 34.23% and 69.24% chance, respectively. The 550 mg/day fixed dose had a probability of attaining the efficacy cutoff comparable to the 10–11 mg/kg daily dose, which had 81.76% chance, but carried also a 41.58% chance of overcoming the tolerability cutoff.

The PTA and CFR values were calculated for 4, 6, 8, 10 and 12 mg/kg daily doses and the results are reported in Figure 6 and Appendix A. Of note, simulations for the 12 mg/kg/day class were at intense risk of bias, because of the paucity of patients treated with the regimen included in the dataset (*n* = 4, see Table 2), and thus are herein omitted. High CFR values (>88%) were obtained with the lowest dosage in MRSA and *S. pneumoniae* infections, while *E. faecium* infections required at least the 8 mg/kg dose to attain a CFR > 83%. For higher MICs, the advantage of the 10 mg/kg daily dose over the 8 mg/kg one was noticeable only in the simulation of PTA for *S. pneumoniae* infections.

## 3. Discussion

Daptomycin is an effective therapeutic option for the treatment of bloodstream infections, infective endocarditis and skin-soft tissues infections sustained by Gram-positive bacteria. Its unique mechanism of action prevents cross-resistances with other antibacterial classes and an inverse resistance phenotype, known as “see-saw effect”, has been described [15]. Nonetheless, several clinical studies [6,7,14] have identified pathophysiological conditions that could alter daptomycin pharmacokinetics and raise the risk of insufficient plasma concentrations. This may finally lead to treatment failures or, worse, to resistant mutants selection [16]. Herein, we evaluated daptomycin pharmacokinetics in a homogeneous population of patients affected by severe Gram-positive infections through the collection of “real-life” TDM results, used to upgrade a previous pop-PK model and then to perform a Monte-Carlo simulation.

Daptomycin plasma concentrations collected in the present study (see Table 2) are in complete agreement with the measures previously made by Falcone et al. [5] and Bubalo et al. [6] in patients affected by severe infections and neutropenic fever, respectively, thus providing further evidence that septic state lowers daptomycin C_max_, as Falcone et al. [7] and Di Paolo et al. [13] already pointed out.

Several daptomycin pop-PK models have been proposed already. At first, there was the original, all-encompassing model by Dvorchik et al. [8], which has been updated in the years by Chaves et al. [9] and Xu et al. [10]. Models proposed by Soraluce et al. [12] and Xie et al. [11] focus on the particular subpopulations of ICU patients and CRRT patients, respectively. Herein, we updated the model initially proposed by Di Paolo et al. [13], which showed more agreement with the lower daptomycin C_max_ dosed in real-life TDM than Dvorchik’s original model [14], but was somehow limited by its small dataset (158 plasma daptomycin concentrations dosed in 58 patients).

Compared to previous parameters estimates, our new model lowers IIV for Cl from 20.74% (with 43.69% relative standard error, RSE) to 2.7% (30% RSE), as well as proportional error, from 36.28% (9.96% RSE) to 29.6% (14% RSE). Goodness-of-fit, compared with previous plots, is substantially improved. Furthermore, our covariate model is not limited to the CrCl effect, but successfully includes sex too, partially “reconciliating” with other models derived from Dvorchik’s one.

Nevertheless, the final covariate model differs from our initial expectations. Firstly, we expected to find a covariate effect of albumin or protein plasma concentration on predicted daptomycin V_d_ or Cl. Despite this, none of the models improved the OFV; thus, they were not retained in the development process. Secondly, we expected to find an effect of weight, instead of sex, on V_d_. Even though it was validated by the least absolute shrinkage and selection operator method [17], the sex-V_d_ relationship conflicts with Dvorchik’s original model, in which weight exerted an effect on the peripheral compartment volume and sex exerted an effect over Cl. In our model, the sex-Cl relationship is already taken into account through the female-adjusting coefficient of the Cockroft-Gault equation. The relationship between sex and V_d_ could be explained mechanistically, since sex-driven differences of adipose tissue abundancy in peripheral compartment, exacerbated in elder patients similar to ours, could alter extravascular distribution of daptomycin. The analysis of the full dataset (*n* = 155) provided us with further insights on the covariate model. Firstly, we graphically recognized a correlation between sex and weight, although it was not statistically significant (data not shown). Considering that in the screening generalized additive model (GAM) for V_d_ sex alone lowered the AIC the most (data not shown), sex may be a proxy for the actual effect of weight. Secondly, in the screening GAM for Cl, the sum of sex, age and plasma albumin lowered the AIC the most (data not shown), and age could be considered a proxy for the creatinine clearance effect. To sum up, we collected preliminary evidence to justify future development of our covariate model towards reconciliation with those derived from Dvorchik’s one and agreement with mechanistic presumptions, through the implementation of creatinine clearance, albumin plasma concentration and sex effect on daptomycin clearance, and weight effect on distribution volume.

Our structural model shares with its precursor an important limitation. Being based on routine TDM, in the framework of an observational retrospective study, only two samples over twenty-four hours were available for each patient. Such data distribution pattern allowed us to embed only a 1-compartment structural model, differently from other pop-PK models based on 2-compartment kinetics. As shown in Figure 2D, this led to imprecisions in the instant concentration estimate between the peak and trough point. Nonetheless, as our parameter estimates (Cl and V_d_ in particular) were in line with the previous literature, we assume that such a limitation did not bias our further model-based simulations.

Indeed, the second aim of the present work was to perform a Monte-Carlo simulation to assess the general tolerability and efficacy of different daptomycin daily dosages. Bhavnani et al. described the probability of CPK elevation fitting an univariate logistic regression on pharmacokinetics data from *S. aureus* BSI patients [18], suggesting that a 24.3 mg/L cutoff for daptomycin trough concentration could be used as a reference for the increased risk of toxicity. According to this reference, our simulated probabilities of the exceeding limit (presented in Table 4 and Figure 5A) are two times greater than the original simulation reported by Di Paolo et al. [14] for the 750mg daily fixed dose, and five to six times greater for other regimens. On the efficacy assessment side, our results substantially agree with previous simulations. As shown in Appendix A, the simulated CFR against MRSA for 666 AUC/MIC target was greater than 95% for the 6, 8 and 10 mg/kg dosages. Compared to the CFR against *E. faecium* reported by Turnidge et al. [19], we found slightly better results at 8 and 10 mg/kg × MIC 2, with 86.3% vs. 80.4% and 99% vs. 92.9%, respectively (see Appendix A). Turnidge’s efficacy assessment was based on simulations made by Avery et al. [20], which in turn applied the Monte-Carlo method to Dvorchik’s pop-PK model. According to the previous literature [6,7,14] and to the present work, their predictions could be, therefore, biased.

In conclusion, herein, we proposed a new pop-PK model for daptomycin, based on the initial observations made by Di Paolo et al. [13], which we validated both in an internal and external manner. We proposed further perspectives of optimization and we applied our model to estimate the general tolerability and efficacy for different daptomycin daily regimens, finding slightly different results from the previous literature. In the near future, we hope that models similar to ours will enter daily clinical practice, helping clinicians to provide patient-tailored therapies through the model-informed precision dosing paradigm.

## 4. Materials and Methods

### 4.1. Patients Enrollment

This work was conducted in the framework of DAPTOLIN, an observational retrospective study approved by the Ethics Committee of the Azienda OspedalieroUniversitaria Pisana (AOUP protocol number 55945). We retrospectively included in the dataset of the present work the patients that fulfilled the following criteria: (1) development of a Gram-positive infection which led to hospitalization in a medical or surgical ward (herein defined as “severe” Gram-positive infection, coherently with [13]); (2) administration of a daptomycin course, ranging from 4 mg/kg to 12 mg/kg, according to the clinical decisions of the ward staff; (3) execution of routine TDM; (4) age ≥ 18 years; (5) signed informed consent for data availability. Patients with incomplete medical records were excluded. As shown in Figure 1, patients were included in two rounds. After the first round, in which 134 patients were included, the dataset was split in 2 subsets through an iterative stratified sampling procedure [21]. The covariates considered for subset balancing were sex, weight, CrCl (calculated through the Cockroft-Gault equation), serum albumin and serum total protein concentration. During the second round, 22 further patients were included, to reach a total of 156 included patients.

### 4.2. Blood Samples

Blood daptomycin concentrations were measured via a routine TDM protocol based on at least two blood samplings for each patient; the “peak” samples were withdrew an hour after the beginning of drug infusion (which most of the times was 30 min long) and the “trough” samples 23 h later. Venous blood samples were collected in heparinized tubes and centrifuged immediately after. The extracted plasma was then stored at −20 °C until subsequent analysis.

### 4.3. Measurement of Daptomycin Plasma Concentrations

The quantification technique was based on the novel HPLC-UV method developed and described by Luci and colleagues [22]. Briefly, plasma samples were firstly purified using a 95:5 *v*/*v* mixture of acetonitrile and 85% H_3_PO_4_, to induce protein precipitation. Samples were then vortexed for 30 s and centrifuged at 7700 rpm for 15 min. The estimated fraction of drug salvaged from the supernatant was 98.2 ± 4.57% (mean ± SD). The mobile phase, composed of a 54:46 *v*/*v* mixture of 20 mM KH_2_PO_4_ and acetonitrile at pH 3.2, was pumped in the HPLC system at a 1 mL/min flowrate, while the chromatographic column temperature was kept at 35 °C. These conditions allowed the optimal separation of analytes in a 10 min-long chromatographic run.

Of note, 5 “trough” concentrations below the limit of quantitation were not included in the modeling dataset.

### 4.4. Pharmacometric Analyses and Model Building

Basic model building was performed using the building subset (*n* = 94), through PsN-controlled NONMEM iterative runs [23]. One- and two- compartment models with first-order elimination, parametrized in terms of drug Cl and Vd, were evaluated. Residual error was described using additive, proportional or mixed error models. Models were compared based on the following aspects: (1) minimization of the OFV; (2) standard errors and confidence intervals of parameter estimates, calculated for each model through NONMEM correlation matrices and PsN-controlled bootstrap method; (3) goodness-of-fit plots generated with Xpose4 [24]; (4) plausibility of parameter estimates, according to the literature.

The preliminary covariate screening was performed through several Xpose4 functions, which are as follows: (1) graphic analysis of covariate-covariate plots to exclude redundancies, (2) graphic analysis of parameters-covariate plots, (3) generalized additive model (GAM) to search for the covariates that minimized the most of Akaike’s information criterion (AIC) value. Covariate model building was based on the least absolute shrinkage and selection operator method, which is automated within the “lasso” Perl command [17]. Further insights on covariates effect were collected with the same methods, after the second patient enrollment round.

### 4.5. Model Validation

The final model was validated both with internal and external approaches. The former one was performed through the following steps: (1) a visual predictive check (VPC), namely a concentration-time plot, where the mean and 5th and 95th percentile of real data are superimposed to the respective descriptive measures of simulated concentrations [25]; (2) a numeric predictive check, which assesses abnormal trends in the DV/iPRED ratio for different statistical confidence intervals; (3) a bootstrap of the model, which calculates the bias between final parameters estimates and the means estimated through 1500 further runs of the same model, based on a random resampling from the same dataset; (4) calculation of *η*- and *ε-shrinkage*, as described in the following equations:(3)$$\eta_{shrink}=1-\frac{stdev\left(\eta_{est}\right)}{\omega }\;$$
(4)$$\varepsilon_{shrink}=1-stdev\left(IWRES\right)\;$$

ω is the population standard deviation of the parameter and *IWRES* is the individual weighted residual, calculated as follows:(5)$$IWRES_{j}=\frac{C_{dv\left(j\right)}-C_{ipred\left(j\right)}}{\sigma }$$

*C_dv(j)_* and *C_ipred(j)_* are, respectively, the measured and individual predicted concentrations of the drug for the *jth* patient, and σ is the standard deviation of residuals, estimated a posteriori from the data.

External validation was conducted running the final model with the validation subset (*n* = 40). Of note, after a preliminary graphic data check, two outliers were found in the series of peak concentrations of the validation subset, which in turn led us to censor those two patients from further computations. Goodness of fit plots and numerical predictive check were executed to evaluate the consistency of the run results.

Finally, individual values of the terminal elimination half-life (t_1/2_) and area under the curve (AUC) were obtained as follows: t_1/2_ (h) = (k_el_)^−1^ and AUC (h × mg/L) = dose/Cl, where k_el_ is the elimination constant.

### 4.6. Model-Based Simulation

Descriptive statistics of the simulation dataset (*n* = 134, see Table 1) were used to simulate in NONMEM 10,000 patients through a Monte-Carlo method. For each daptomycin fixed and weight-adjusted dosage, two probabilities were assessed. The first was to overcome a C_min_ of 24.3 mg/L, as a toxicity threshold in agreement with Bhavani et al. [18]. Efficacy was preliminary investigated by calculating the probability of reaching maximum plasma concentrations (C_max_) of >60 mg/L, and a more precise assessment was based on the probability of target attainment (*PTA*) and the cumulative fraction of response (*CFR*), calculated as follows [26]:(6)$$CFR=\overset{n}{\underset{i=1}{\sum }}PTA_{i}\times F_{i}$$

*F* stands for the MIC-stratified observation fraction on total bacterial population observations. EUCAST’s MIC distributions for MRSA, *E. faecalis* and *S. pneumoniae* were used to calculate the predicted AUC/MIC values. In particular, the AUC/MIC target values were 666 for MRSA [14], 438 for *S. pneumoniae* [27] and 294.45 for *E. faecium* (resulting from the correction for mean albumin binding of the 27.4 *f*AUC/MIC target) [19].

### 4.7. Other Statistical Analyses

Data are presented as mean ± SD (or median and 95% confidence interval (95% CI) when appropriate). Student’s t-test was used to check for significant differences between groups after the execution of the subsampling algorithm. All calculations were performed with Microsoft Excel 365 (Microsoft, Redmond, WA, USA) and R 3.6.2.

## Figures and Tables

**Figure 1 antibiotics-11-00914-f001:**
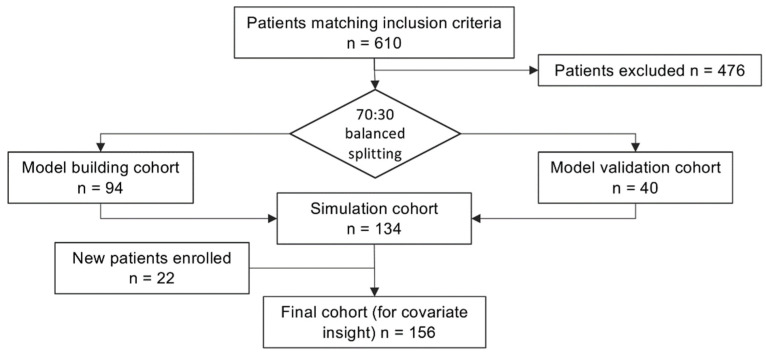
Flow diagram showing the patient inclusion process of the present study.

**Figure 2 antibiotics-11-00914-f002:**
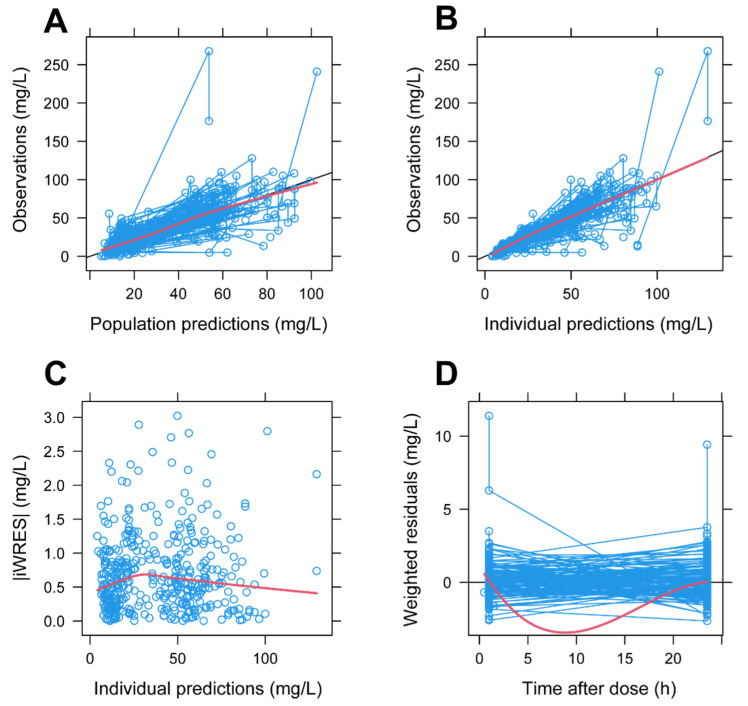
Goodness-of-fit plots of the final model: (**A**) relationship between observations and population prediction of daptomycin plasma concentrations; (**B**) relationship between observations and individual prediction of plasma concentrations; (**C**) distribution of absolute values of individual weighted residuals (|iWRES|) versus individual predictions of daptomycin plasma concentrations; (**D**) distribution of weighted residuals versus time after dose. Blue lines and empty circles: daptomycin plasma concentrations for every enrolled patient. Red lines: mean of the observed concentrations (panels (**A**,**B**)) or Loess lines (panels (**C**,**D**)). Black lines: lines of identity.

**Figure 3 antibiotics-11-00914-f003:**
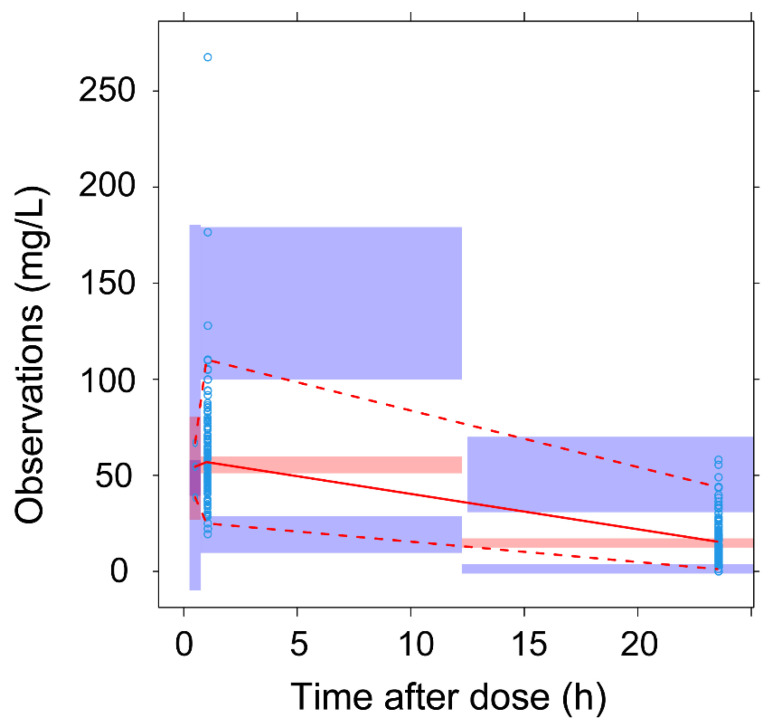
Visual predictive check (VPC) of the final model. Empty circles: observed data; red solid line: median of observed data; red dashed lines: 5th and 95th percentiles of observed data; pink area: 95% confidence interval around the median of simulation; purple areas: 95% confidence interval around 5th and 95th percentiles of simulation.

**Figure 4 antibiotics-11-00914-f004:**
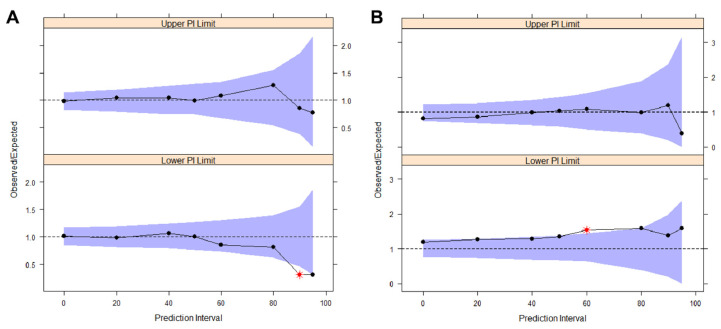
Numerical predictive check (NPC) coverage plots for the final model: (**A**) internal validation NPC, run on the train dataset; (**B**) external validation NPC, run on the test dataset. Solid circles: mean DV/iPRED ratios calculated on 1000 resampling from the dataset. Purple areas: theoretical intervals of the DV/iPRED ratios for the different statistical confidences (on the *x*-axis).

**Figure 5 antibiotics-11-00914-f005:**
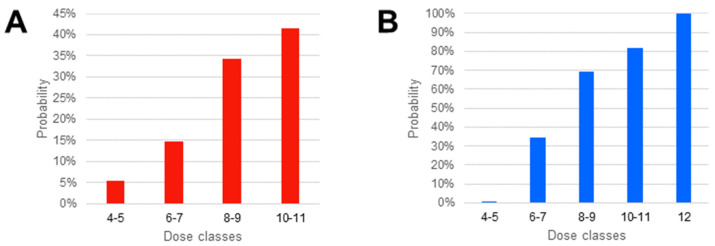
Analysis of predicted concentrations in 10,000 patients simulated through Monte-Carlo method: (**A**) probability of attaining a C_min_ > 24.3 mg/L, as tolerability index; (**B**) probability of attaining a C_max_ > 60 mg/L, as preliminary efficacy index.

**Figure 6 antibiotics-11-00914-f006:**
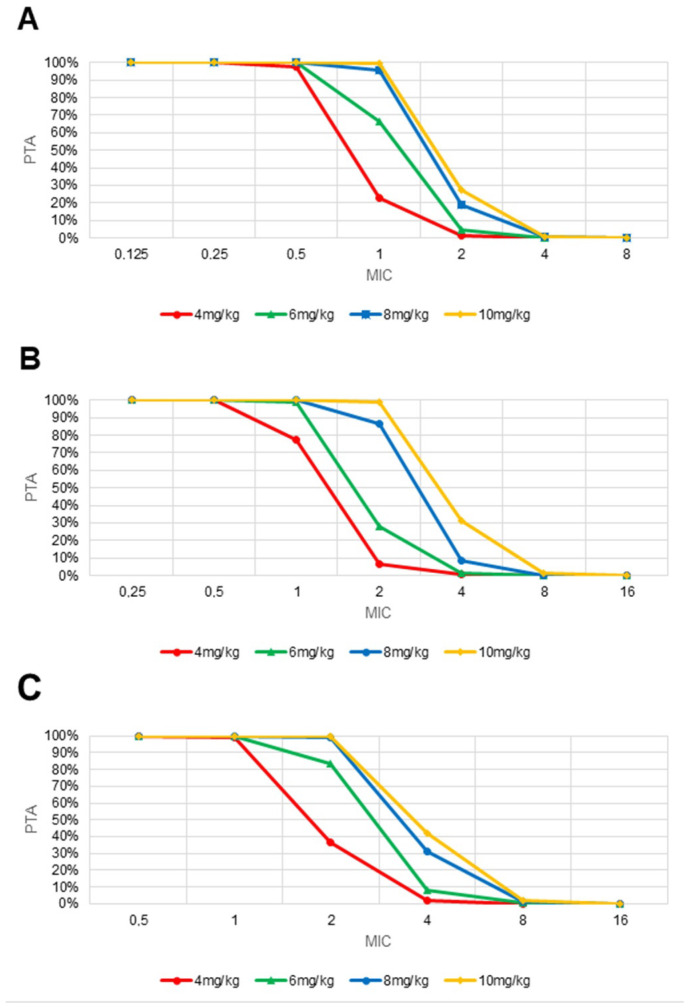
Probability of target attainment (PTA) in (**A**) MRSA, (**B**) *S. pneumoniae* and (**C**) *E. faecium* infections, according to minimum inhibitory concentration (MIC) values ranging from 0.25 up to 16 mg/L. Simulated daily doses: 4 mg/kg (red), 6 mg/kg (green), 8 mg/kg (blue), and 10 mg/kg (yellow). Data for the 12 mg/kg daily doses were uninformative, and thus omitted.

**Table 1 antibiotics-11-00914-t001:** Synthesis of patients’ data included in the present study, grouped by cohort.

	Building	Validation	Simulation	Covariate Insights
Patients (n)	94	40	134	156
Male (n)	65	25	90	108
Female (n)	29	15	44	48
Age (years)	65.7 ± 13.2	66.1 ± 18.0	65.8 ± 14.7	66.0 ± 14.0
Weight (kg)	72.6 ± 10.9	73.8 ± 10.0	72.9 ± 10.6	73.9 ± 11.2
CrCl (mL/min/1.73 m^2^)	74.6 ± 39.7	74.5 ± 41.0	74.6 ± 39.9	73.6 ± 38.3
Serum albumin (mg/dL)	3.2 ± 0.6	3.3 ± 0.5	3.2 ± 0.6	3.3 ± 0.6
Serum proteins (mg/dL)	6.7 ± 1.0	6.5 ± 0.8	6.6 ± 0.9	6.7 ± 0.9
Daptomycin				
Total daily dose (mg)	491.2 ± 115.8	498.8 ± 112.4	493.4 ± 114.4	516.5 ± 131.9
Daily dose (mg/kg)	6.8 ± 1.6	6.8 ± 1.6	6.8 ± 1.6	6.7 ± 1.5

**Table 2 antibiotics-11-00914-t002:** Values (mean ± SD, minimum–maximum range) of 424 daptomycin plasma concentrations measured in the present study.

Daptomycin Daily Dose (mg/kg)	Patients (n)	Daptomycin Plasma Concentrations			
		1 h		23.5 h	
		Mean ± SD	Range (Min–Max)	Mean ± SD	Range (Min–Max)
4–5	15	45.6 ± 17.1	18.1–87.6	11.4 ± 8.7	BLQ–32.6
6–7	96	59.3 ± 18.0	19.4–110.0	17.8 ± 11.0	BLQ–55.5
8–9	32	74.4 ± 51.3	BLQ–268.0	15.5 ± 10.7	BLQ–44.6
10–11	8	69.8 ± 21.7	33.6–105.0	25.4 ± 14.1	6.6–58.1
12	4	80.0 ± 25.7	49.3–108.0	17.3 ± 9.2	17.3–29.8

Abbreviations: BLQ, below the limit of quantitation.

**Table 3 antibiotics-11-00914-t003:** Final parameter estimates of the pop-PK model (left) and mean bootstrap estimates (right).

Parameters		Final Model			Bootstrap	
		Mean	SE	RSE (%)	Mean	95% CI
**Cl (L/h)**	θ_1_	0.636	0.037	6	0.631	0.598–0.674
**V (L)**	θ_2_	10.925	0.414	4	10.935	10.291–11.559
**Additive Error (mg/L)**	θ_3_	3.805	1.610	42	3.841	2.319–5.290
**Proportional Error (%)**	θ_4_	0.296	0.038	14	0.266	0.219–0.320
**k_CrCl_**	θ_5_	0.109	0.179	164	0.128	0.012–0.207
**k_Sex_**	θ_6_	−2.524	0.941	37	−2.484	−3.840–1.208
**Error Slope for Higher iPRED**	θ_7_	−0.546	0.266	49	−0.293	−1.312–0.220
**IIV_Cl_**	*η*	0.027	0.008	30	0.024	0.018–0.036

Abbreviations: Cl, clearance; V, volume of distribution, k_CrCl_, constant for creatinine clearance effect on Cl; k_Sex_, constant for sex effect on V_d_; IIV_Cl_, interindividual variability in drug clearance; SE, standard error; RSE, relative standard error; 95% CI, 95% confidence intervals.

**Table 4 antibiotics-11-00914-t004:** Probability of overcoming the tolerability and efficacy cut-off values (24.3 mg/L and 60 mg/L, respectively), calculated on the 10,000 simulated patients.

	Probability	
Fixed Dose (mg/day)	C_min_ > 24.3 mg/mL	C_max_ > 60 mg/L
300	2.40%	0.70%
350	6.20%	3.60%
400	8.70%	9.70%
450	10.50%	25.70%
500	16.05%	30.40%
550	21.50%	78.60%
750	44.10%	100%
Simulation Total	20.44%	40.80%

## Data Availability

A limited access to the data will be available upon request according to local procedures and protocols.

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
