# Peer review of "Daptomycin Population Pharmacokinetics in Patients Affected by Severe Gram-Positive Infections: An Update"

_antibiotics, 2022, doi:10.3390/antibiotics11070914_

Round 1
Reviewer 1 Report
In the present manuscript, authors have characterized Daptomycin PK using a pop-PK modeling approach.
- The English language usage, typos and grammatical errors needs to be thoroughly corrected throughout the manuscipt.
- In the model described here, what is different and novel than the model previously published by Di Paolo et al and how does this update the already existing knowledge about Daptomycin pharmacokinetic variability and the factors impacting that.
Reviewer 2 Report
Title: Daptomycin population pharmacokinetics in patients affected by severe Gram-positive infections: an update
General Comments:
Authors investigated the population pharmacokinetics of daptomycin in a larger group of patients who receive the drug to treat severe Gram-positive infections like endocarditis, and the model was applied to perform a Monte-Carlo simulation and assess tolerability and efficacy of different daptomycin dosages.
Authors results may help clinicians to choose a patient-tailored daptomycin dose in order to maximize the odds of therapy success without neglecting toxicity risk.
Manuscript written well and this research could be of interest for the field.
Reviewer 3 Report
The manuscript analyzed the plasma concentrations of daptomycin in patients with severe Gram-positive infections, reported population PK modeling analysis of these data and their implications. Unfortunately, there are substantial flaws in data analysis that undermine the scientific merit of this manuscript.
Specific comments:
1. Inclusion criteria (line 288) – how severe Gram-positive infection was defined?
2. Daptomycin generally follows 2-compartment PK, as reported e.g, in ref 8 (Dvorchick et al., 2004; several previous manuscripts that reported PopPK analysis of daptomycin are not cited in the paper: Xie et al, J Antimicrob Chemother 2020; 75: 1559–156; Xu et al., Br J Clin Pharmacol (2017) 83 498–509). 2 datapoints per patient, at 2 specific pre-set timepoints, are insufficient to depict the time course of the drug concentrations over the studied period (0-24 hr).
3. It is not clear how many data points were defined as outliers, and how they were defined (Grubbs test?). It is not clear how many BLQ samples were collected, and how these data were handled during the modeling analysis.
4. Analysis of covariates (eq. 1 & 2) is non-standard, and appears to be inappropriate. What is the meaning of the constants in these equations (63.35, 0.309)? The claims regarding the low IIV of CL (line 25) are inappropriate, in view of the observed data (Fig. 2) and high IIV of KCrCL (table 3). Overall, daptomycin has substantial/high IIV, especially in the patients with severe disease (see e.g., Xie et al, J Antimicrob Chemother 2020; 75: 1559–156).
5. The breakpoint & safety were defined based on Cmax & Cmin (lines 367-371). This appears to be a heavily biased way to predict the drug effects, as daptomycin AUC/MIC ratio is expected to correlate the best with the in vivo efficacy of this drug (see e.g., Xie et al, J Antimicrob Chemother 2020; 75: 1559–156)
6. Extensive English language proofing is needed to correct numerous terminological and grammatical errors.
Round 2
Reviewer 3 Report
The authors made substantial changes to the manuscript, according to the specific points of critique. Despite these changes, I feel that the applied PopPK modeling analysis in inappropriate, in terms of 1) insufficient experimental data (2 data points only for a drug that follows 2-compartmental PK), 2) the applied covariate model (eq. 1 & 2, a substantially different model as compared to the commonly used-model, and to that used by Maximova et al 2022), 3) selection of the safety and efficacy thresholds based in Cmin & Cmax.
